# Pre-Therapeutic Assessment of Older People in Sub-Saharan Africa: Introduction to the Comprehensive Geriatric Assessment

**DOI:** 10.3390/jcm13061801

**Published:** 2024-03-21

**Authors:** Marie-Josiane Ntsama Essomba, Berthe Sabine Esson Mapoko, Junette Arlette Metogo Mbengono, Nadine Simo-Tabue, Andre Pascal Kengne, Simeon Pierre Choukem, Eugène Sobngwi, Jacqueline Ze Minkande, Maturin Tabue Teguo

**Affiliations:** 1Department of Internal Medicine and Specialties, Faculty of Medicine and Biomedical Sciences, University of Yaounde I, Yaounde P.O. Box 812, Cameroon; ebomaj2012@yahoo.fr (M.-J.N.E.); mapokob@yahoo.fr (B.S.E.M.); sobngwieugene@yahoo.fr (E.S.); 2Department of Surgery and Specialties, Faculty of Medicine and Pharmaceutical Sciences, University of Douala, Douala P.O. Box 2701, Cameroon; junetmell@yahoo.fr; 3Pôle de Geriatrie/Gérontologie CHU de Martinique, Equipe EpiCliV, Université des Antilles, 97233 Fort-de-France-Martinique, France; nadine_tabue@yahoo.fr; 4Non-Communicable Diseases Research Unit, South African Medical Research Council, University of Cape Town, Cape Town 7700, South Africa; andre.kengne@mrc.ac.za; 5Department of Biological and Environmental Sciences, Faculty of Natural Sciences, Walter Sisulu University, Mthatha 5700, South Africa; 6Department of Internal Medicine and Specialties, Faculty of Medicine and Pharmaceutical Sciences, University of Dschang, Dschang P.O. Box 96, Cameroon; schoukem@gmail.com; 7Department of Surgery and Specialties, Faculty of Medicine and Biomedical Sciences, University of Yaounde I, Yaounde P.O Box 812, Cameroon; minkandeze@yahoo.fr

**Keywords:** CGA model, multidisciplinary team, geriatric surgery, cancer

## Abstract

**Objectives**: With the ongoing epidemiological transition in sub-Saharan Africa (SSA), conditions that require invasive treatment (surgery, cancer, and anaesthesia, etc.) will become increasingly common. Comprehensive geriatric assessment (CGA) is a multidisciplinary diagnostic process aimed at identifying older people at risk of negative outcomes. It is important to know whether this approach integrates care management strategies for older people in a context where health services for older people are scarce, and staff members have little training in geriatrics. The current work is a situational analysis on the use of CGA on invasive care (cancer, surgery, etc.) among older people in SSA. **Methods:** We searched PubMed-MEDLINE and other sources for studies reporting on CGA and conditions requiring invasive treatment in older patients in SSA. **Results/Conclusions:** We found no study that had comprehensively examined CGA and invasive care in SSA. There is, however, evidence that the offer of invasive care to older people has improved in SSA. Further research is needed to explore the applicability of CGA in SSA. Similarly, more investigations are needed on the role of CGA in the care trajectories of older people in SSA, in terms of outcomes and affordability.

## 1. Introduction

In 2020, people aged 65 and over have outnumbered children younger than 5 years worldwide [1]. According to the World Health Organization (WHO), nearly 70% of people aged 60 and over currently live in developing countries, and this proportion is expected to increase further [2]. With the ongoing epidemiological transition in sub-Saharan Africa (SSA), there is a rising trend in non-communicable diseases (NCDs), including conditions requiring surgical procedures and anesthetic care [3,4]. The prevalence of unmet surgical need for older people in Uganda was 27.8% in 2014 [5]. The same year in Ghana, the annual operation rate of people aged 65 and over was 1744 per 100,000 [6]. Although surgical techniques and anesthesia have globally improved during the last decade in SSA, ageing will bring about new challenges. Older surgical patients are prone to postoperative complications, functional decline, prolonged recovery, and increased mortality. Indeed, older people have a higher risk of poor postoperative outcomes, owing to age-related changes, multimorbidity, and frailty [7,8,9]. Furthermore, in the presence of multimorbidity, coordination among multiple specialties can be difficult, and there is less focus on interventions to improve long-term outcomes after surgery. Surgeons and anesthesiologists must consider the balance between the benefits of a potentially life-sustaining surgical procedure versus the risks of developing postoperative disabling complications in older patients. Existing preoperative risk assessment tools are focused on optimizing immediate postoperative care and do not fully predict an older patient’s capacity to withstand the surgical risk [10]. Older people usually present with physical problems but also with functional, psychological, and social issues. In a traditional approach, the standard of care has been mainly focused on treating the disease and has failed to embrace the complex and multifactorial nature of these conditions. In the early 1980s, Rubenstein et al. provided a comprehensive interdisciplinary evaluation and management of frail older patients at high risk of long-term institutionalization following acute hospitalization [11]. Comprehensive geriatric assessment (CGA) is a multidisciplinary model of care intended to identify frail older patient’s medical, functional, and psychosocial limitations to develop a global plan of treatment and long-term follow-up [12]. Rather than looking only at diseases as occurs in routine medical assessments, CGA involves looking at a range of domains including physical medical conditions, mental health, functioning, and social circumstances. These assessments across multiple domains are required to develop a broad therapeutic plan to enhance recovery and promote the independence of patients. It is, therefore, both a diagnostic and a therapeutic process. This model is designed to reduce the incidence and worsening of functional disability of hospitalized older adults in the context of an acute illness [13,14]. It has been argued that combining usual risk assessment tools and CGA in frail older patients could improve the identification of patients who are likely to exhibit negative outcomes after older patient care [7,15]. Little consideration has been given to issues of aging in SSA, yet several lines of evidence support the importance of the equitable distribution of the available healthcare resources across age groups. Furthermore, out-of-pocket contribution remains the main source of healthcare expenses. But it is not enough, especially when dealing with complex health problems of older adults. In a context where health services dedicated to older people are scarce and few personnel are trained in geriatrics, we aimed to explore the literature on the use of CGA in the management of invasive care for older patients in SSA.

Source of the evidence: 

A systematic Medline literature search was conducted on July 2023 without limit of date to identify all articles examining the CGA in SSA. The literature search was performed across mainly PubMed-Medline and other sources (Science Direct, Google Scholar) using Medical Subject Heading (MeSH) terms with the following search strategy: “aging” OR “older” OR “Africa” combined with MeSH terms “surgery” OR “cancer” OR “frailty” OR “CGA” OR “hip fracture” OR “orthogeriatric” OR “oncogeriatric”. To optimize the searches, we used the same terms in the Title/Abstract (TiAb) advanced search option. Screening via title, abstract, and full-text, where appropriate, identified no study on the theme of interest from SSA. 

Accordingly, we present in this paper a narrative synthesis of the elements underlying CGA’s interest in strategies for caring for older people. 

### Geriatric Syndromes and Population Aging in SSA

According to census data and population projections, about 160 million older people will be living in Africa by 2050. The older population is expected to double in the majority of African countries between 2020 and 2050, and these countries will probably need a policy shift to meet the needs of this aging population [16,17,18]. In SSA, life expectancy at the age of 60 years is 16 years for women and 14 years for men, suggesting that for those who survived the challenges in early age, a long old age is a reality [19]. Health systems in SSA are still struggling with the heavy burden of infectious diseases while facing the rising trend of NCDs [3,20]. Indeed, the increasing burden of many chronic diseases including hypertension, diabetes, but also cancer and osteoporosis, is a matter of concern in SSA. Older people usually present with physical problems but also with functional, psychological, and social issues. Furthermore, there are some clinical conditions known as geriatric syndromes, highly prevalent in older people living with frailty that do not fit into discrete disease categories [21]. Geriatric syndromes are multifactorial and associated with substantial morbidity and poor outcomes [22,23]. The main geriatric syndromes include frailty, falls, delirium, cognitive impairment, functional decline, and urinary incontinence [21]. Despite efforts to improve the level of care, mortality remains high in this age group [24,25,26] owing to comorbidities, but the role of geriatric syndromes has also been recognized in SSA [27,28]. The prevalence of frailty has been reported in many SSA countries and varies widely depending of the method of assessment [8,27,29]. According to a systematic review conducted in 2021 by O’Caoimh et al., the highest physical frailty prevalence was reported in Africa compared to other regions of the world [30]. In Nigeria, 63% of hospitalized adults aged 60 years and above were living with frailty [27] while in community-dwellers aged 55 years and above in Cameroon, frailty has been reported in 36% of participants [31]. In a study conducted in an urban setting in 2019, geriatric syndromes were highly prevalent (67%) among Cameroonians aged 55 and above; these included activities of daily living dependency and cognitive impairment in 10% and 30% of cases, respectively [32]. There is also a growing interest on sarcopenia, a geriatric condition characterized by the progressive and generalized loss of muscle mass and strength, associated with high risk of poor outcomes [33]. Indeed, its prevalence in SSA varies between 5 and 53% depending on the diagnostic criteria [34,35,36,37] (Table 1).

## 2. Surgical Conditions in Older People in SSA

With the ongoing epidemiological transition in sub-Saharan Africa (SSA), there is a rising trend in NCDs including conditions requiring surgical procedures and anesthetic care [3,4]. Several older people undergo surgical procedures in SSA, reaching 46% of all admissions in some surgery units [39,40]. People aged 65 years and above accounted for about 31% of admissions in a surgery ward in Ethiopia [41]. The prevalence of the unmet surgical need of older people in Uganda was 27.8% in 2014 [5]. During the same year in Ghana, the annual operation rate of people aged 65 and over was 1744 per 100,000 [6]. It has long been recognized that advanced age can increase the risk of poor outcomes during and after surgery. Older people usually present with a background of complex medical issues including comorbidities but also age-related conditions. Furthermore, a substantial role of neighborhood, social, and community factors in seeking and reaching injury care has been highlighted [42]. A clinical audit conducted in Nigeria found a crude mortality rate of 18.9% in people aged 65 years and above in the general surgery unit [43]. In Cote d’Ivoire, post-operative morbidity and mortality among older people undergoing emergency surgery for acute bowel obstructions were 32.2% and 16.9%, respectively [44]. In a follow-up study including 2530 participants, the hazard of postoperative mortality among older people was three-folds higher than among their younger counterparts in Ethiopia [45]. The reasons of this high mortality were not clearly reported, as pre-existing medical conditions were not investigated in the majority of these studies. It has been suggested that older people face delayed diagnosis and late referral owing to limited access to health facilities and financial issues [5,6,46,47]. However, several authors have highlighted the increasing prevalence of geriatric syndromes in various settings in SSA [8,27,29,32]. Undiagnosed pre-existing cognitive dysfunction was present in about 40% of older elective non-cardiac surgery patients in South Africa [48]. Despite the existing geriatric syndromes and the widespread knowledge that older patients of the same age do not all have the same risk of occurrence of adverse outcomes, reliable data on geriatric assessment in surgery are lacking in many African countries. Several conditions in older people can require surgical management. Surgical techniques and anesthesia have globally improved during the last decade in SSA. Furthermore, the medical diaspora being actively engaged has contributed to this improvement through collaborative partnerships and transfer of skills. The CGA responds to the need to adapt care where necessary, and the specific needs of the older person are taken into account. This assessment can be used as a basis for treatment in cardiology, neurology, pulmonology, etc. Nevertheless, oncology and surgery are two models in which CGA has been used to demonstrate its effectiveness. For the purpose of this narrative review, we focus on cancer and fragility hip fracture. 

### 2.1. Fragility Hip Fractures

The burden of injuries is a global challenge among older people especially, fragility hip fractures (FHFs), owing to their increased risk of falling and high prevalence of osteoporosis. In a study conducted in South Africa, the FHF incidence reported was 19.3 per 100,000, reaching 23.4 in the female group [49], and it was associated with an average of 16 days’ hospital stay following surgery [50]. Falls account for the major causes of FHF in older patients in SSA, often in combination with other geriatric syndromes such as sarcopenia and frailty [8,39,47,51,52]. In Malawi, orthopedic surgery was the most common procedure performed in older patients admitted for trauma with a longer hospital stay and higher mortality [47]. The 1-year mortality rate after hemiarthroplasty was around 34% in South Africa [53]. FHF is an acute stress condition in older people living with frailty, with increased risk of fatal adverse events as well as considerable economic burden [54,55]. In FHF patients, mortality ranges from 8 to 36% within the first year after surgery, and excess annual mortality persists for more than 10 years thereafter [56]. Multimorbidity and geriatric syndromes present difficulties for orthopedic surgeons, leading to multiple medical consultations, thus delaying surgery, extending the length of stay, and finally increasing mortality. To improve outcomes and reduce hospital costs, the full integration of orthopedics and geriatrics in dedicated units has been implemented [57]. Orthogeriatric co-management is a model of care which consists of a systematic collaboration between orthopedic surgeons, geriatricians, and the multidisciplinary geriatric team [58,59]. Consistent meta-analysis has shown that the orthogeriatric co-management of older patients with FHF significantly reduces long-term mortality as well as length of stay [58,59,60]. Despite the burden associated with FHF in older patients in SSA, orthogeriatric co-management is not widely implemented mainly because there are few geriatricians in SSA.

### 2.2. Cancer

Both cancer cases and cancer-related mortality have increased significantly in Africa [20,61]. Aging is one of the strongest risk factors for cancer development, but cancer screening coverage remains low among older people in SSA [62]. Despite the scarcity of population-based data on the cancer burden in the older African population, prostate cancer, breast cancer, and cervical cancer are frequently reported [63,64,65,66,67]. At least in high-income countries, the prevalence of frailty is high amongst older patients with cancer [68]. Both cancer and cancer treatments are significant stressors that can challenge the physiological reserve; thus, one should note the importance of an early assessment. An integrated oncogeriatric approach provides coordinated healthcare to address both cancer and age-related needs. Actually, several frailty screening tools are available, but the Geriatric 8 (G8) is the most widely used to identify older cancer patients who can benefit from further assessment and appropriate care [69,70]. Several randomized controlled trials suggest that CGA improves outcomes such as quality of life, unplanned hospitalizations, and chemotherapy completion [71,72]. Furthermore, CGA increases communication about age-related concerns and reduced treatment toxicity [73]. Surgery is an essential component for global cancer care in all resource settings. It is often the only curative option for many solid tumors. Detailed information concerning surgical oncology in the African continent is rudimentary. 

## 3. Comprehensive Geriatric Assessment: A Cornerstone of Modern Geriatric Care

Comprehensive geriatric assessment (CGA) is defined as a multidimensional interdisciplinary diagnostic process focused on determining frail older patient’s medical, functional, and psychosocial capability in order to develop a coordinated and integrated plan of treatment and long-term follow-up [12,13]. Rather than looking only at diseases as standard medical assessment would do, CGA involves looking at a range of domains including the following: physical medical conditions, mental health, functioning, and social circumstances [13,14]. CGA is delivered according to two broad models: in the first one, patients are admitted in a dedicated ward, usually called Acute Care for Elders (ACE) model with a coordinated multidisciplinary team who perform both assessment and rehabilitation [74,75]. The second model consists of a mobile team who visits patients living with frailty wherever they are admitted in non-geriatric wards. The mobile team assesses the patients and makes recommendations to the physician in charge [76]. The CGA is formalized by the use of standardized scales and tools, although they can be clinically constraining for non-trained healthcare providers (Table 2). These assessments across multiple domains are required to develop a broad therapeutic plan to enhance recovery and promote the independence of patients. It is, therefore, both a diagnostic and a therapeutic process. CGA has been confirmed by high-quality research studies and subsequent meta-analyses as the gold standard for the in-hospital care of geriatric patients [13,15,77]. This model is designed to reduce the incidence and worsening of functional disability of hospitalized older adults in the context of an acute illness [13,14]. A better identification of vulnerable patients also facilitates shared decision-making between patients, families, and caregivers about realistic treatment goals. The interest in frailty is growing with regard to efforts to increase healthy living expectancy and improving care to older people across various settings. Furthermore, the WHO has recently published guidelines to promote integrated care for older people in the community (ICOPE) [78]. ICOPE provides tools and guidance to community health workers to detect the decline in intrinsic capacities, including impaired mobility, malnutrition, visual impairment, hearing loss, cognitive impairment, and depressive symptoms and to deliver interventions for their management [78,79,80,81]. Given the demonstrated benefits from CGA for older people living with frailty, it is important that this model of care is more widely understood especially in the African setting [82].

### 3.1. Perioperative Optimisation of Older People

Clinical evidence shows that CGA reduces postoperative medical complications and is cost effective in the perioperative setting [15]. With the growing recognition of the need to tailor services to the pathophysiological profile of older people, there is an increasing interest in the Perioperative Optimization of Older People undergoing elective and urgent surgery (POPS). POPS aims to assist older people in choosing whether surgery is the right option, optimizing physiology before surgery, and helping to manage postoperative complications and plan rehabilitation. It can be employed throughout the whole perioperative pathway, allowing for the anticipation and mitigation of postoperative complications and proactive discharge planning [83]. In a traditional approach, CGA is usually delivered by a geriatrician-led multidisciplinary team. Although POPS is a CGA-based model, the scarcity of geriatricians in SSA is a major barrier to the widespread implementation of POPS services for older people. 

### 3.2. Future Perspective

Several pieces of evidence support the implementation of CGA-based interventions for older people living with frailty, but reliable data on CGA in daily practice are lacking in many African countries. There may be a number of reasons that explain the underuse of CGA in SSA. CGA is time- and resource-consuming, which makes it difficult to put in place for surgeons and anesthesiologists usually focused on optimizing immediate postoperative care. Furthermore, geriatric medicine is a new specialty with few trained health caregivers and scarce dedicated wards [84]. Nonetheless, recent data suggest that it is possible to establish a model of care to address older people’s health issues in resource-limited settings [85]. In Cameroon, the Acute Care for Elders model has been implemented in the lone geriatric-dedicated unit of the country since 2019 [28]. A dedicated team has been put in place including geriatricians, general practitioners, advanced practice nurses, and social workers who provide geriatric care and CGA for all hospitalized patients. Their role has been expanded to assess outpatients and older patients admitted in non-geriatric wards. Although the evidence for its use in hospitals remains strong, further work is needed to explore the applicability of CGA in SSA. Perioperative medicine is being established to provide optimal preoperative, intraoperative, and postoperative care for all patients, but with a particular focus on those at high risk of adverse postoperative outcomes. Education and research are keys in expanding geriatric care to currently underserved areas in SSA. This can be conducted locally through web-based conferences and short workshops, as well as partnership with well-established geriatric departments abroad. In this process, programmes to promote primary prevention in the community are not only relevant, but also necessary in the African setting. The ICOPE approach can serve as a template for further national guidelines, taking into account the available healthcare resources. To achieve this goal, commitment is needed from all relevant stakeholders including governments, researchers, and healthcare professionals in addition to civil society actors. 

## 4. Conclusions

Although surgical techniques and anesthesia have globally improved during the last decade in SSA, ageing will bring about new challenges. Evidence for the existing geriatric syndromes is available, but reliable data on geriatric assessment in invasive care are lacking in many African countries. CGA provides a contrasting model of care to the traditional approach, and its benefits are evidence-based in various settings. Further research is needed to fill the gap on whether this approach can be suitable and implemented in sub-Saharan Africa, taking into account local resources. 

## Figures and Tables

**Table 1 jcm-13-01801-t001:** Prevalence of frailty in sub-Saharan Africa.

Author, Year	Setting	Country	Sample Size Mean or Median Age % Female Study Design	Frailty Measure	Prevalence	Association between Frailty and Adverse Outcome
Leopold-George et al. [8], 2016	In-hospital surgical ward	South Africa	299 50.6 52% Prospective cohort study	Clinical frailty scale	22.4%	Desaturation (OR 4.21; *p* = 0.01), Blood transfusion requierment (OR 5.36, *p* = 0.01) Higher ASA-PS scores (OR 19.01, *p* < 0.001)
Adebusoye et al. [27], 2019	In-hospital medical ward	Nigeria	450 71.5 52% Prospective cohort study	Canadian Study of Health and Aging (CSHA) clinical frailty scale	63.3%	The 30-day all-cause mortality rate was significantly higher among frail respondents (18.8 deaths per 1000 patient days compared with non-frail respondents—11.3 deaths per 1000 patient days)
Metanmo et al. [31], 2023	Community	Cameroon	403 67.1 49.6% Cross-sectional study	Study of osteoporotic fractures index	35.7%	NA
Payne et al. [29], 2017	Community	South Africa	5059 61.7 54.1% Longitudinal cohort study	Fried’s criteria	5.4% to 13.2%	The 17-month all-cause mortality HR 2.65 to 8.91 for frail vs. non-frail
Witham et al. [38], 2019	Community	Burkina Faso	2973 54 50.6% Cross-sectional study	Fried’s criteria	7%	Frailty was strongly associated with impairment of activities of daily living and with lower wealth, being widowed, diabetes mellitus, hypertension, and self-reported diagnoses of tuberculosis or heart disease

ASA-PS The American Society of Anaesthesiologists Physical Status; OR odds ratio; HR hazard ratio.

**Table 2 jcm-13-01801-t002:** Components of CGA.

Domain	Assessment	Example of Tools
Physical medical conditions	Comorbid conditions and disease severity Medication review	CIRS-G, Charlson’s index
Functioning	Core functions such as mobility and balance Activities of daily living	ADL IADL
Cognition	Cognition Mood and anxiety	MMSE GDS-15
Nutrition	Nutritional status	MNA
Social network	Social networks: informal support available from family, the wider network of friends and contacts, and statutory care Poverty	UCLA-loneliness Scale
Environment	Housing: comfort, facilities, and safety Use or potential use of “telehealth” Technology Transport facilities Accessibility to local resources	

ADL: Activities of Daily Living; CGA: comprehensive geriatric assessment; CIRS-G: Cumulative Illness Rating Scale for Geriatrics;. GDS-15: Geriatric Depression Scale 15; IADL: Instrumental Activities Of Daily Living; MMSE: Mini–Mental State Exam; MNA: Mini Nutritional Assessment.

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
