# Peer review of "Pre-Therapeutic Assessment of Older People in Sub-Saharan Africa: Introduction to the Comprehensive Geriatric Assessment"

_jcm, 2024, doi:10.3390/jcm13061801_

Round 1
Reviewer 1 Report (Previous Reviewer 1)
Comments and Suggestions for Authors
The authors have addressed the points raised by my previous review - thank you
Reference 42 is not the correct reference by Owolabi - it should be Owolabi et al, Ann Glob Health 2023; 89: 5 (the one currently quoted deals with a different topic)
Comments on the Quality of English Language
The manuscript would benefit from an edit by a native English speaker - for example "evidences" is not a correct term in English.
Author Response
Dear reviewers,
We acknowledge that you spent a substantial amount of time looking over the paper that shows the interest of this topic. Thank you for all your relevant comments and suggestions.
Reviewer 1
Comments and Suggestions for Authors
The authors have addressed the points raised by my previous review - thank you.
Reference 42 is not the correct reference by Owolabi - it should be Owolabi et al, Ann Glob Health 2023; 89: 5 (the one currently quoted deals with a different topic)
Our answer: We have checked the reference. The correct reference 42 have been inserted: (1)1. Owolabi EO, Ferreira K, Nyamathe S, Ignatowicz A, Odland ML, Abdul-Latif A-M, et al. Social Determinants of Seeking and Reaching Injury Care in South Africa: A Community-Based Qualitative Study. Ann Glob Heal [Internet]. 2023 Jan 27;89(1). Available from: https://annalsofglobalhealth.org/articles/10.5334/aogh.4003/
Comments on the Quality of English Language:
The manuscript would benefit from an edit by a native English speaker - for example "evidences" is not a correct term in English.
Our answer: We have reviewed the manuscript again and fixed typos and grammatical issues identified.

Reviewer 2 Report (New Reviewer)
Comments and Suggestions for Authors
An interesting review article has been produced to help assess the role of comprehensive geriatric assessment in the care of older people in sub-Saharan Africa.
However, it has identified a number of shortcomings that should be addressed.
Introduction: it should be explained in more depth what a comprehensive geriatric assessment is, what it involves, what it is for and what benefits it can bring.
Material and method: this section is missing, it is nowhere to be found. It is necessary to add the search procedure. Inclusion and exclusion criteria and a flow chart (CONSORT method) should be added. How many articles were finally selected? It is not clear how many articles were considered for the review.
I would ask to add a table with the articles that were finally selected, extracting the characteristics of each of them.
Discussion: it is necessary to add a well-defined discussion section in which the results obtained from the search carried out are commented on.
This review article is not well thought out initially, as it is not clear how it is organised or what the final aim of the article is.
Author Response
Comments and Suggestions for Authors
An interesting review article has been produced to help assess the role of comprehensive geriatric assessment in the care of older people in sub-Saharan Africa.
However, it has identified a number of shortcomings that should be addressed.
Introduction: it should be explained in more depth what a comprehensive geriatric assessment is, what it involves, what it is for and what benefits it can bring.
Our answer: Thank you for this comment, we have added more details on CGA’s line 24-34
Material and method: this section is missing, it is nowhere to be found. It is necessary to add the search procedure. Inclusion and exclusion criteria and a flow chart (CONSORT method) should be added. How many articles were finally selected? It is not clear how many articles were considered for the review.
I would ask to add a table with the articles that were finally selected, extracting the characteristics of each of them.
Discussion: it is necessary to add a well-defined discussion section in which the results obtained from the search carried out are commented on.
This review article is not well thought out initially, as it is not clear how it is organised or what the final aim of the article is.
Our answer:
CGA in the management of serious diseases such as orthogeriatric, oncogeriatric, etc. is known to have a positive impact on the prognosis of older adults in middle- and high-income countries. We thought it appropriate to take stock of this approach to the care of the elderly in SSA, given the projected ageing of the population. We aimed to present the evidence that is already available on how CGA is used in invasive care in sub-Saharan Africa. Although publications on ageing in African countries are increasing, we found no articles on CGA and invasive care in SSA. We hope that this narrative review will pave the way for further research into whether this approach can be adapted and/or implemented in sub-Saharan Africa, taking into account local resources.
As suggested, our review was carried out according to the Preferred Reporting Items for Systematic Reviews and Meta-analysis (PRISMA) guidelines. As the protocol does not exactly follow the format accepted in PROSPERO®, it is not included in the PROSPERO® registry.

Reviewer 3 Report (New Reviewer)
Comments and Suggestions for Authors
Title: Pre-therapeutic Assessment of Older People in Sub-Saharan Africa: introduction to the Comprehensive Geriatric Assessment
The topic of this article is interesting and relevant in showing the pre-therapeutic assessment of older people in Sub-Saharan Africa. Several references are also quoted to support the implementation of the Comprehensive Geriatric Assessment (CGA) interventions for older people living with frailty. Although reliable data on the CGA in daily practice are lacking in many African countries, the usefulness of the CGA in Africa should not be underestimated. Therefore, the work is a significant contribution to the field.
The objectives of the article are stated clearly. The methods used in this literature review were applicable and explained well. The conclusion is clear and reflects the objectives of the manuscript. The content of the article is well organized and comprehensively described. The references are appropriate and adequate to related and previous work. The English used is correct and readable.
However, there are some minor corrections to be done.
1) Abstract: Ensure that the following headings are clear and in sequence order in your abstract: Introduction, Objectives of the review, Methods, Results, Discussion, Conclusion and Recommendation.
2) Ensure that the abovementioned headings are also clear in your manuscript.
3) Nearly 20 of your References are older than 5 years. Please update where possible
4) Reference 24: Change date to bold.
Author Response
The topic of this article is interesting and relevant in showing the pre-therapeutic assessment of older people in Sub-Saharan Africa. Several references are also quoted to support the implementation of the Comprehensive Geriatric Assessment (CGA) interventions for older people living with frailty. Although reliable data on the CGA in daily practice are lacking in many African countries, the usefulness of the CGA in Africa should not be underestimated. Therefore, the work is a significant contribution to the field.
The objectives of the article are stated clearly. The methods used in this literature review were applicable and explained well. The conclusion is clear and reflects the objectives of the manuscript. The content of the article is well organized and comprehensively described. The references are appropriate and adequate to related and previous work. The English used is correct and readable.
However, there are some minor corrections to be done.
- Abstract: Ensure that the following headings are clear and in sequence order in your abstract: Introduction, Objectives of the review, Methods, Results, Discussion, Conclusion and Recommendation.
Our answer: The abstract has been adapted according to the writing instructions to authors.
2) Ensure that the abovementioned headings are also clear in your manuscript.
3) Nearly 20 of your References are older than 5 years. Please update where possible
Our answer: change implemented where possible
4) Reference 24: Change date to bold.
Our answer: we have checked the reference. change implemented

Reviewer 4 Report (New Reviewer)
Comments and Suggestions for Authors
The article "Pre-therapeutic Assessment of Older People in Sub-Saharan Africa: Introduction to the Comprehensive Geriatric Assessment" presents several limitations.
Although the study is described as a systematic review, many essential elements (practically all) required for a systematic review are not presented. The authors should consult the PRISMA guidelines and reconsider their research.
Regarding the few points presented:
-the Methodology should be presented as a separate section of the article and not as part of the introduction;
-the Review mentions the use of Medline, but subsequently, other databases are cited;
-it is unclear whether the search terms were consistent across each database.
The authors state that the study's objective is "to review the literature on the use of CGA on invasive care (cancer, surgery, etc.) among older people in SSA." However, they describe (page 3) that
"No study on this subject is available from SSA. Accordingly, we present in this paper the elements underlying CGA's interest in strategies for caring for older people."
As a result, we do not have a review that addresses what was mentioned in the abstract. In fact, it is a poorly conducted narrative review that mostly discusses disjointed points without adequate coverage.
Thus, considering the poorly formulated and unanswered objective, the disjointed structure of the text, the failure to follow systematic review methods, and the presentation of results that do not allow for establishing conclusions regarding the formulated hypothesis, unfortunately this work does not constitute a scientifically robust product.
Comments on the Quality of English LanguageSome minor issues.
Author Response
Although the study is described as a systematic review, many essential elements (practically all) required for a systematic review are not presented. The authors should consult the PRISMA guidelines and reconsider their research.
Regarding the few points presented:
-the Methodology should be presented as a separate section of the article and not as part of the introduction;
-the Review mentions the use of Medline, but subsequently, other databases are cited;
-it is unclear whether the search terms were consistent across each database.
The authors state that the study's objective is "to review the literature on the use of CGA on invasive care (cancer, surgery, etc.) among older people in SSA." However, they describe (page 3) that "No study on this subject is available from SSA. Accordingly, we present in this paper the elements underlying CGA's interest in strategies for caring for older people."
As a result, we do not have a review that addresses what was mentioned in the abstract. In fact, it is a poorly conducted narrative review that mostly discusses disjointed points without adequate coverage.
Thus, considering the poorly formulated and unanswered objective, the disjointed structure of the text, the failure to follow systematic review methods, and the presentation of results that do not allow for establishing conclusions regarding the formulated hypothesis, unfortunately this work does not constitute a scientifically robust product.
Our answer: CGA in the management of serious diseases such as orthogeriatrics, oncogeriatrics, etc. is known to have a positive impact on the prognosis of older adults in middle- and high-income countries. We thought it appropriate to take stock of this approach to the care of the elderly in SSA, given the projected ageing of the population. We aimed to present the evidence that is already available on how CGA is used in invasive care in sub-Saharan Africa. While conducting this narrative review, we found no articles on CGA and invasive care in SSA. Indeed, despite the increasing number of publications on invasive care, especially surgery in older patients in SSA, relatively few having assessed geriatric syndromes. Further research is required to assess the use of CGA and as we were conducting this review, none have comprehensively examined this model of care in African countries. We hope that this narrative review will pave the way for further research to fill the gap on whether this approach can be suitable and/or implemented in sub-Saharan Africa, taking into account local resources.
As suggested, our review was carried out according to the Preferred Reporting Items for Systematic Reviews and Meta-analysis (PRISMA) guidelines. As the protocol does not exactly follow the format accepted in PROSPERO®, it is not included in the PROSPERO® registry.

Round 2
Reviewer 2 Report (New Reviewer)
Comments and Suggestions for Authors
A proper review of the comments previously made has been made.
A good review of the existing literature has been made.
Author Response
Thank you for the reviewer
Reviewer 4 Report (New Reviewer)
Comments and Suggestions for Authors
In their response, the authors made some adjustments in the text.
However, they did not provide sufficient justification or corrections for the majority of the presented limitations. The altered study still does not adhere to PRISMA recommendations.
The authors have now labeled their review as a narrative, but this does not mitigate the shortcomings in methods, results organization, and discussion.
The authors' efforts to address some concerns are appreciated; however, the study still lacks the necessary robustness, whether classified as a systematic or narrative review, particularly in light of the unresolved limitations and non-compliance with PRISMA guidelines.
Author Response
We thank you for re-emphasizing these points. We are fully aware of the PRISMA guidelines as collectively, investigators on this paper have co-authored over 200 systematic reviews (with or without meta-analysis). The systematic review approach to the current paper was futile as no relevant studies were found. We therefore opted to offer a reflection on the elements underlying CGA’s interest in strategies for caring for older people, with a focus on Africa. It is exactly what we said in the ‘source of evidence’ section which was added at the request of the reviewer. Strictly speaking, this is more an ‘opinion leaders’ type paper, for which the PRISMA guidelines are not applicable.
This manuscript is a resubmission of an earlier submission. The following is a list of the peer review reports and author responses from that submission.
Round 1
Reviewer 1 Report
Comments and Suggestions for Authors
This short review is a helpful introduction to an important, but understudied topic in global ageing. I have the following suggestions to further enhance the manuscript:
Abstract:
As a narrative review, the methods section of the abstract is probably the right length, but I would like to see a bit more in the results/conclusions section. Highlight the key findings and also where the key gaps are in the literature that require further work – be specific. If necessary, reduce the length of the objectives section of the abstract to stay within the word limits
Line 23: should be etc rather than ie after surgery, cancer, anaesthesia
Line 24: Older patients, not olders patients
Introduction:
No comments
Section 1.1:
Please change all instance of ‘frail older people’ (and similar phrases) to ‘older people living with frailty’
Line 90: Change to “Geriatric Syndromes and population aging in sub-saharan Africa” (referring to SSA provides consistency throughout the article)
Lines 91-99 would fit better in the introduction section
There is a recent systematic review reporting the prevalence of frailty in a large number of countries, subgrouped by continent: This would be worth quoting in the section on frailty: O’Caoimh R et al. Age Ageing 2021;50:96-104
Section 1.2
It is perhaps worth drawing attention to the wide range of healthcare systems and infrastructure across SSA – many factors contribute to surgical outcomes – for instance, late presentation to first responders, long transfer distances from home to hospital, quality of available surgical services, as well as quality of supporting services. All of these factors may impact differently on older surgical patients, but they may differ greatly – what happens to an older person in urban South Africa will for instance be very different to what happens to an older person in rural Burkina Faso. One recent paper that explore some of these factors (for seeking injury care) is Owolabi et al, Ann Glob Health 2023; 89: 5 (although age is not a key focus of this particular analysis)
Section 1.3
Avoid using the word ‘elderly’ – use ‘older’ or ‘older people’
It would be helpful to add a few lines to explain why orthogeriatric co-management is not widely implemented – clearly the main reason is that there are very few geriatricians in SSA, but this is worth spelling out.
Section 1.4
You state that frailty is common in patients with cancer – this is certainly true in high-income countries (and the reference you quote is from high-income countries). Are there any data on the prevalence of frailty in LMICs? If not, it would be important to say that, and to say “At least in high-income countries, the prevalence of frailty is high amongst older patients with cancer”
Section 1.5
The key section I would like to see added is a section discussing the specific evidence for CGA in surgical patients. You have already mentioned orthogeriatric and oncogeriatric services, but there is now a growing literature on perioperative optimisation of older patients (POPS) – helping older people choose whether surgery is the right option (it is not always), optimising physiology before surgery, and helping to manage postoperative complication and deliver rehabilitation. Whilst this is based on CGA, it is a distinct enough model to merit discussion in its own right. See for example:
Partridge JSL, Moonesinghe SR, Lees N, Dhesi JK. Perioperative care for older people. Age Ageing. 2022 Aug 2;51(8):afac194.
Partridge JS, Harari D, Martin FC, Peacock JL, Bell R, Mohammed A, Dhesi JK. Randomized clinical trial of comprehensive geriatric assessment and optimization in vascular surgery. Br J Surg. 2017 May;104(6):679-687.
Partridge JS, Harari D, Martin FC, Dhesi JK. The impact of pre-operative comprehensive geriatric assessment on postoperative outcomes in older patients undergoing scheduled surgery: a systematic review. Anaesthesia. 2014 Jan;69 Suppl 1:8-16.
Furthermore, it would be helpful to discuss how such POPS models could be adapted for delivery in resource-constrained settings such as those found in SSA. This to me seems to be where the rest of the paper is leading and would greatly strengthen the ending of the paper.
References:
Check references – a few appear to have the authors first name listed instead of surname (e.g. references 28 and 34)
The manuscript would benefit from editing to optimise English grammar.
Comments on the Quality of English LanguageSee above
Reviewer 2 Report
Comments and Suggestions for Authors
The work presents extremely relevant topics for the treatment of older people suffering from serious illnesses. However, although it shows that it wants to identify the evaluative parameters used in geriatric management, the research question is not clear. Another important point to be noted is that the method has low replicability potential, therefore, I suggest applying the PICO search strategy, and better outlining the search procedures adopted. The presented structure is framed as an integrative one, deliberating low power of evidence related to the data presented.